# Minimally Invasive Surgery in Non-Small Cell Lung Cancer: Where Do We Stand?

**DOI:** 10.3390/cancers15174281

**Published:** 2023-08-26

**Authors:** Lawek Berzenji, Wen Wen, Stijn Verleden, Erik Claes, Suresh Krishan Yogeswaran, Patrick Lauwers, Paul Van Schil, Jeroen M. H. Hendriks

**Affiliations:** 1Department of Thoracic and Vascular Surgery, University of Antwerp, 2610 Wilrijk, Belgium; 2Antwerp Surgical Training, Anatomy and Research Centre (ASTARC), Laboratory of Thoracic and Vascular Surgery, 2650 Edegem, Belgium; 3Department of Thoracic and Vascular Surgery, Antwerp University Hospital, Drie Eikenstraat 655, 2650 Edegem, Belgium

**Keywords:** minimally invasive surgery, video-assisted thoracoscopic surgery, robotic-assisted thoracoscopic surgery, non-small cell lung cancer, lobectomy, sublobar resections, nodule detection

## Abstract

**Simple Summary:**

In the last twenty years, minimally invasive surgery (MIS) has radically changed the surgical landscape. In the field of thoracic surgery, approaches such as video-assisted thoracoscopic surgery (VATS) and robotic-assisted thoracoscopic surgery (RATS) have become the new standards for the majority of procedures performed, especially for early-stage lung cancer. Despite these developments, there is still a lack of concrete data regarding treatment outcomes of these minimally invasive approaches compared to the conventional open surgery. In the future, the number of minimally invasive procedures will likely keep increasing as more lung cancer nodules are detected at early stages due to lung cancer screening initiatives. Therefore, data on short- and long-term outcomes of VATS and RATS in early-stage lung cancer is needed.

**Abstract:**

In the last two decades, robotic-assisted thoracoscopic surgery (RATS) has gained popularity as a minimally invasive surgical (MIS) alternative to multi- and uniportal video-assisted thoracoscopic surgery (VATS). With this approach, the surgeon obviates the known drawbacks of conventional MIS, such as the reduced in-depth perception, hand-eye coordination, and freedom of motion of the instruments. Previous studies have shown that a robotic approach for operable lung cancer has treatment outcomes comparable to other MIS techniques such as multi-and uniportal VATS, but with less blood loss, a lower conversion rate to open surgery, better lymph node dissection rates, and improved ergonomics for the surgeon. The thoracic surgeon of the future is expected to perform more complex procedures. More patients will enter a multimodal treatment scheme making surgery more difficult due to severe inflammation. Furthermore, due to lung cancer screening programs, the number of patients presenting with operable smaller lung nodules in the periphery of the lung will increase. This, combined with the fact that segmentectomy is becoming an increasingly popular treatment for small peripheral lung lesions, indicates that the future thoracic surgeons need to have profound knowledge of segmental resections. New imaging techniques will help them to locate these lesions and to achieve a complete oncologic resection. Current robotic techniques exist to help the thoracic surgeon overcome these challenges. In this review, an update of the latest MIS approaches and nodule detection techniques will be given.

## 1. Introduction

Lung cancer is the most common malignant tumour globally and a leading cause of cancer-related deaths [1,2,3,4,5,6]. While rates vary across countries, there has been an overall increase in new cases of non-small cell lung cancer (NSCLC), particularly in developing nations [7]. GLOBOCAN estimates for 2020 showed 2.2 million new lung cancer diagnoses, accounting for 11.4% of all new cancer diagnoses, and 1.8 million deaths due to lung cancer, representing 18.0% of all cancer-related deaths [8]. Unfortunately, the five-year survival rate for lung cancer remains around 18%, much lower than other leading cancers [9]. To improve outcomes, lung cancer screening initiatives using low-dose computed tomography (CT) scans have been proposed, with trials like NELSON and NLST demonstrating reduced mortality rates in high-risk groups by detecting nodules at earlier stages [10,11,12].

Minimally invasive surgery (MIS) has revolutionized surgical practice worldwide, combining technological advancements like high-definition cameras and micro-instruments to enable complex procedures through small incisions [13]. MIS has allowed for safe and feasible laparoscopic and thoracoscopic procedures for many diseases, resulting in fewer complications, shorter hospital stays, and faster recovery times compared to open surgeries [14,15]. In thoracic surgery, video-assisted thoracoscopic surgery (VATS) has shown fewer perioperative complications, less pain, and faster recoveries than an open approach using a thoracotomy [16]. However, proficiency in MIS requires extensive training with steep learning curves for surgical residents. Surgeons may face challenges with poor depth perception, diminished spatial coordination due to two-dimensional optics, a lack of instrument flexibility, and counter-intuitive movements [17,18]. Furthermore, surgeons are often exposed to physical strain from operating in uncomfortable positions for extended periods, which can increase surgical complexity and impact patient outcomes.

In the last two decades, robotic-assisted surgery has emerged as a new minimal-invasive approach, combining the latest technological advancements with conventional MIS. Robotic-assisted surgery involves the use of a robotic device to control endoscopic instruments through a remote console, providing the surgeon with three-dimensional optics, a wider range of instrument motions, and improved ergonomics [15,19]. For thoracic surgery, robotic-assisted thoracoscopic surgery (RATS) has been shown to offer significant benefits, though there are still controversies surrounding RATS, such as high operating costs and longer procedure times due to the installation of the robotic device [20,21]. Furthermore, there is limited data from large randomized controlled trials (RCTs) comparing short-term and long-term outcomes of RATS and VATS. Such data are crucial to demonstrate the superiority of RATS in terms of morbidity, mortality, postoperative recovery, cost-effectiveness, and long-term safety.

Currently, the gold standard for early-stage non-small cell lung cancer (NSCLC) is surgical management via lobectomy with hilar and mediastinal lymph node dissection. [22,23]. However, emerging evidence has indicated that sublobar resections may be equal to lobectomy in selected cases [24,25]. These findings, combined with future lung cancer screening initiatives, will drastically change the landscape for lung cancer treatments worldwide. For the thoracic surgeon, this means an increase in patients that are eligible for segmentectomy, a significantly more complex procedure than lobectomy, in addition to more challenging localizations of nodules due to their smaller sizes. Techniques such as radio-guided localization, microcoil/hookwire placement, intraoperative ultrasonography, fluorescence-guided lung nodule identification, and navigational bronchoscopy will be useful tools for surgeons in these cases [26,27,28,29]. This review aims to provide a summary of the latest data regarding surgical techniques and treatment outcomes for MIS in the management of lung cancer.

## 2. VATS/RATS Lobectomy

In 1995, the Lung Cancer Study Group (LCSG) published their landmark RCT in which patients with peripheral stage T1N0 NSCLC tumours were randomly assigned perioperatively to undergo either lobectomy or sublobar resection (segmentectomy or wedge resection). In this prospective multicentre trial, 247 eligible patients were included. The difference in overall survival (OS) favoured the lobectomy group but did not reach statistical significance. However, the incidence rate of locoregional recurrence was three times higher in the sublobar resection group compared to the lobectomy group. Following this trial, lobectomy was set as the gold standard of surgical care for this group of patients, and sublobar resections were reserved for patients with limited pulmonary function [22]. Despite the scarcity of well-powered RCTs comparing RATS and VATS with open surgery in the treatment of early-stage lung cancer, several studies have indicated that lobectomy via either VATS or RATS offers benefits compared to open surgery. These benefits include reduced intraoperative blood loss, shorter hospital length of stay (LOS), decreased need for postoperative pain management, improved postoperative recovery, and lower 30-day mortality rates [30,31,32,33]. Recently, results of the VIOLET trial were published, in which 503 patients were randomly assigned to VATS (*n* = 247) or open (*n* = 256) lobectomy. The primary outcome was physical function at 5 weeks using the European Organisation for Research and Treatment of Cancer (EORTC) core health-related quality of life questionnaire (QLQ-C30). The results showed that median physical function was significantly better in the VATS arm compared to the open surgery arm (73 versus 67, respectively, with a mean difference of 4.65 points [95% confidence interval (CI), 1.69–7.61]). Furthermore, patients had shorter postoperative hospital LOS despite more air leaks and bleeding, fewer SAEs after discharge and readmissions, and less pain. After 1 year, no differences were observed regarding cancer progression-free survival (PFS) (hazard ratio, 0.74 [0.43–1.27]) or OS (hazard ratio, 0.67 [0.32–1.40]) [34].

Previous research, such as the 2014 Nationwide Inpatient Sample study, indicated that RATS had a greater risk of cardiovascular complications and iatrogenic bleeding compared to VATS [35]. However, more recent studies have not demonstrated any significant differences in short-term outcomes between VATS and RATS [33,36]. In a 2018 meta-analysis by Liang et al., early outcomes of RATS and VATS were compared in 3239 patients. The authors found lower 30-day mortality and numbers of conversion to open thoracotomy in favour of RATS [37]. In a meta-analysis that included 3375 subjects for RATS and 58,683 subjects for VATS, Emmert et al. showed an improved survival for RATS for patients undergoing lung resections for all types of diseases. In a propensity-matched analysis of RATS versus open lobectomy (*n* = 2775 each) and RATS versus VATS lobectomy (*n* = 2951 each), Oh et al. demonstrated lower postoperative complication rates and shorter hospital stays for RATS compared to VATS. However, no difference in mortality rate between VATS and RATS was demonstrated [38,39]. In a similar database study by the Society of Thoracic Surgeons (STS) which included 1220 RATS and 12,378 VATS lobectomies, no significant difference between the two approaches was found regarding OS [40]. More recently, Zhang et al. performed a meta-analysis of 26 studies including a total of 45,773 patients (14,271 RATS and 31,462 VATS procedures). No significant differences were found regarding operative time, complications, tumour size, chest tube duration, R0 resection rate, lymph stations sampled, 5-year OS, and recurrence rate. However, the RATS group had significantly less perioperative blood loss, lower conversion rates to open surgery, shorter hospital LOS, improved lymph node dissection, and better 5-year disease-free survival (DFS) [41]. 

Several studies in the last few years have attempted to evaluate oncological outcomes of RATS surgery compared to open surgery or VATS. Wilson et al. demonstrated a higher rate of nodal upstaging in patients undergoing RATS compared to VATS or open surgery in their retrospective study of 302 patients [42]. Similar findings were shown by Yang et al. regarding nodal station sampling with RATS [43]. However, a database analysis of the National Cancer Database (NCDB, which included 64,676 patients from the USA and compared RATS to VATS and open thoracotomy, failed to demonstrate improved lymph node yield or nodal upstaging of NSCLC in RATS compared to VATS or open surgery [44]. The authors concluded that, based on their data analysis, both RATS and conventional VATS are non-inferior to open surgery for perioperative lymph node evaluation. Although the majority of these studies show promising results, RCTs are necessary to effectively demonstrate the superiority of MIS over open surgery, especially for oncological outcomes. Recently, the results of the ROMAN study, a prospective international RCT, were published, in which perioperative outcomes and surgical radicality of RATS were compared to VATS for early-stage NSCLC (cT1-2; N0-1). The primary objective of this trial was the incidence of adverse events (AEs). The trial was closed at 83 patients as the observed trends showed that the probability of concluding in favour of the RATS arm was null. However, there was a significant improvement in lymph node sampling for the RATS arm [45]. Another recent trial was the RVlob Trial by Jin et al., a single-centre, open-labelled prospective RCT which aimed to compare the efficacy of lobectomy by RATS and VATS. A total of 320 patients were enrolled and randomly assigned to either lobectomy by RATS (*n* = 157) or VATS (*n* = 163). Perioperative outcomes were similar in both groups, however, lymph node yield and positive N1 nodes were significantly higher in the RATS group [46]. Currently, the RAVAL trial, an ongoing international, multicentre RCT trial is investigating outcomes of RATS versus VATS lobectomy [47]. The preliminary results were presented at the American Association for Thoracic Surgery meeting in 2022. The early results showed that RATS lobectomy is a cost-effective treatment associated with better patient-reported health-related quality of life (QoL). Furthermore, significantly more lymph nodes were sampled in the RATS arm. The long-term oncological results will be analysed in later phases of the trial. Table 1 shows an overview of the most recent large meta-analyses and RCTs comparing VATS and RATS lobectomy.

In recent years, uniportal VATS (UniVATS) has gained popularity as an approach with the aim of reducing surgical trauma during thoracic surgery. In this approach, the thoracic surgeon performs his procedure using one incision and specific instruments designed for UniVATS. Although this approach has been described in some reports already dating back more than 20 years ago, it is only quite recently that studies have shown its safety and feasibility in lung cancer surgery [52]. Furthermore, evidence suggests that, in terms of postoperative pain, UniVATS is likely to be better, or at least no worse, than the standard three-port VATS approach [53,54]. The European Society of Thoracic Surgeons (ESTS) has set up a Uniportal VATS Interest Group (UVIG) with the aim of encouraging research into outcomes of UniVATS. In their consensus paper published in 2019, the UVIG Consensus Report stated that UniVATS offers a valid alternative to standard VATS techniques [55]. Although a number of smaller studies have compared UniVATS to standard VATS, only very limited data is available regarding outcomes of UniVATS versus RATS. In a propensity score-matched analysis by Yang et al., a total of 153 patients treated with UniVATS (*n* = 77) or RATS (*n* = 76) were retrospectively analysed for postoperative outcomes. The authors concluded that UniVATS and RATS were similar regarding operative time, postoperative hospital LOS, chest tube duration, use of analgesia, complications, or number of resected lymph nodes. However, RATS was associated with less intraoperative blood loss and more dissected lymph node stations [43]. 

Even for more complex cases and locally advanced cancers, RATS has shown to be feasible and effective in several studies [56,57]. Although no large trials have been published regarding RATS for locally advanced NSCLC, a number of retrospective studies have shown favourable results. In a retrospective, multicentre study by Veronesi et al., perioperative outcomes, recurrence rates, and OS were analysed in a total of 232 patients with evident or occult N2 disease (210 NSCLC and 13 carcinoid tumours). Their results showed that 98.4% of all patients had R0 margins with a conversion rate to thoracotomy of 9.9%. Furthermore, 23 patients (10.3%) had serious (grade III-IV) postoperative complications. The authors concluded that RATS lobectomy is safe and effective in patients with stage III NSCLC or carcinoid tumours [58]. In another study, Herb et al. performed a retrospective analysis of the NCDB of patients that underwent lobectomy (open, VATS, or RATS) for stage IIIA-N2 NSCLC from 2010 to 2016. A total of 5741 patients were identified (3879 open, 1403 VATS, and 459 RATS). VATS and RATS both had lower 90-day and 5-year mortality rates compared to open resections. Furthermore, among the MIS approaches, RATs had a better 90-day mortality rate compared to VATS [59]. Recently, a similar analysis of the NCDB was performed by Baig et al., in which patients were included with NSCLC and either clinical N1/N2 disease or a tumour > 5 cm. All patients had received neoadjuvant chemotherapy/radiotherapy and were treated with VATS or RATS. A total of 9512 patients (2123 RATS and 8389 VATS) were identified. No significant differences were found regarding R) resections, 30- and 90-day mortality rates, or 30-day readmission rates. However, VATS had a significant higher conversion rates to thoracotomy compared to RATS [60]. Some recent data has shown that even sleeve lobectomies using a RATS-approach is feasible and has favourable outcomes [61]. In a recent single-centre retrospective study by Liu et al., 104 patients with centrally located NSCLC underwent RATS sleeve lobectomy. The authors reported 5-year DFS and OS rates of 67.9% and 73.0%, respectively. They concluded that RATS lobectomy could be an oncologically adequate procedure for patients with centrally located NSCLC [62].

We are entering a new era of thoracic surgery with new approaches and techniques being developed. In the near future, uniportal RATS (UniRATS) will likely become a popular surgical approach as well. Preliminary studies have already shown the safety and feasibility of UniRATS in lung cancer surgery [63,64]. Larger randomized studies are needed to corroborate these early findings. However, the future thoracic surgeon will not only have to take into account new developments in surgical approaches, but also developments in postoperative care. There is discussion regarding optimal pain management after lung cancer surgery, resulting in many different guidelines and a lack of consensus [65]. The majority of patients receive thoracic epidural analgesia (TEA) following VATS [66]. However, data suggests that continuous paravertebral block or single-shot intercostal nerve block are effective alternatives to TEA [67]. Currently, the OPtriAL is investigating the postoperative pain outcomes of these three analgesia modalities after VATS lung resections [68]. In addition to these new approaches and techniques, new guidelines for indications for surgical resections will become necessary in this era of immunotherapy and targeted therapy. Very little is known regarding outcomes of robotic-assisted surgery in patients that have received immunotherapy. However, preliminary data from smaller studies are promising. In a recent real-world single-centre prospective cohort study by Gao et al., a total of 44 patients who underwent RATS after three doses of neoadjuvant chemoimmunotherapy were included for analysis. In this group, 36 (81.8%) patients had a major pathological response and 26 (59.1%) had a pathological complete response. R0 resection was achieved in all patients. Two (4.5%) patients required conversion to thoracotomy. The authors concluded that RATS after neoadjuvant chemoimmunotherapy showed good feasibility and safety in stage III NSCLC [69]. Nevertheless, robust data on surgical outcomes of induction immunotherapy is still lacking and further studies are necessary to demonstrate its efficacy. 

Despite all the data from the trials and studies mentioned above, it is important to put these findings into perspective. The costs of implementing these techniques and approaches in addition to technical or organizational difficulties should be balanced against the possible benefits. Oncological outcomes and long-term OS for open surgery are similar to MIS, despite benefits in short-term results for MIS such as postoperative pain and complication rates. Furthermore, data on the completeness of lymph node dissection and staging for MIS, especially VATS, remains unclear and warrants further investigations [70]. In addition to the issue of long-term outcomes, the learning curve and the training required for performing VATS and RATS should also be considered [71,72]. Moreover, training programs for future thoracic surgeons should aim to find a balance between teaching trainees skills in MIS and open surgery for cases in which conversion to open surgery is necessary or in cases of fiscal crises and instrument shortages [73,74]. In addition to this, patient-reported outcomes comparing VATS and RATS are necessary as well. Although the majority of studies do not show large differences in patient outcomes, many of them report conflicting findings regarding postoperative pain and QoL. Further studies comparing these two approaches are necessary to accurately measure patient-reported outcomes. 

## 3. Segmentectomy for Early-Stage Lung Cancer

Since the landmark trial by Ginsberg et al., lobectomy with systematic mediastinal lymph node dissection has been accepted as the standard of care for early-stage NSCLC [75,76]. Sublobar resections for early-stage NSCLC has only been indicated for selected patients with poor pulmonary function or with other comorbidities prohibiting lobectomy. However, lobectomy results in significant loss of lung tissue and diminished QoL compared to sublobar resections. Sublobar resections, such as wedge resection and anatomical segmentectomy, have been suggested as alternative surgical techniques for elderly patients or those with limited pulmonary reserve [76,77]. Furthermore, with the increase in CT-screening programmes and advances in in diagnostic modalities, a trend can be seen of early detection of small-sized nodules and ground-glass opacities (GGO). Consequently, experts are advocating to extend the indications of sublobar resections to early-stage NSCLC [78]. Currently, two recent trials have attempted to provide an answer to the question whether lobar and sublobar resections are equal with regard to treatment outcomes. In the Japanese multicentre open-label phase-III non-inferiority trial (JCOG0802/WJOG4607L), 1106 patients with clinical stage IA NSCLC were enrolled and randomly assigned to receive lobectomy (*n* = 554) or segmentectomy (*n* = 552). Baseline clinicopathological factors were balanced between the two patient groups. Their findings showed a 5-year OS of 94.3% [95% CI, 92.1–96.0%] for segmentectomy and 91.1% [95% CI, 88.4–93.2%] for lobectomy, with a median follow-up of 7.3 years. Statistical analyses of their data showed that this difference in OS was significant (*p* < 0.0001 for non-inferiority and *p* = 0.0082 for superiority). Five-year relapse free survival was 88.0% [95% CI 85.0–90.4%] for segmentectomy and 87.9% [95% CI 84.8–90.3%] for lobectomy. Locoregional relapse rates were almost twice as high in the segmentectomy group compared to lobectomy (10.5% vs. 5.4%, respectively). However, no significant difference was noted for the 5-year relapse free survival (*p* = 0.9889). The authors hypothesize that, perhaps due to preservation of more lung parenchyma, patients in the segmentectomy group were more likely to receive more extensive treatments for (local) relapses or for other malignant or non-malignant diseases, resulting in a higher OS rate. In addition, a significant difference in reduction in forced expiratory volume in 1 s of 3.5% was found as well, favouring the segmentectomy group. However, this difference did not reach the predefined clinical threshold of 10% in this study. No differences in postoperative complication rates were found. The authors concluded that segmentectomy, rather than lobectomy, should be the standard surgical procedure for patients with small, peripheral NSCLC lesions (≤2 cm, consolidation-to-tumour ratio > 0.5) [79]. Shortly after this trial, Altorki et al. presented the results of their multicentre, phase III, non-inferiority trial, which included a total of 697 patients with peripheral T1aN0 NSCLC to be randomly assigned to undergo sublobar resection (wedge resection or segmentectomy; *n* = 340) or lobectomy (*n* = 357). After a median follow-up of 7 years, sublobar resection was non-inferior to lobectomy for DFS (HR for disease recurrence or death, 1.01; 90% CI, 0.83–1.24). In addition, OS after sublobar resection was similar to that after lobectomy (HR for death, 0.95; 95% CI, 0.72–1.26). The 5-year DFS was 63.6% (95% CI, 57.9–68.8) after sublobar resection and 64.1% (95% CI, 58.5–69.0) after lobectomy. The 5-year OS was 80.3% (95% CI, 75.5–84.3) after sublobar resection and 78.9% (95% CI, 74.1–82.9) after lobectomy. No significant differences were noted between the two arms regarding the incidence of locoregional or distant recurrence. The authors concluded that sublobar resections for patients with clinical T1aN0 disease is non-inferior to lobectomy [80]. Table 2 shows an overview of ongoing RCTs regarding segmentectomy for NSCLC.

While there is an increasing amount of data suggesting the value of segmentectomy as an alternative to lobectomy in certain cases, the long-term outcomes of robotic segmentectomy remain unclear. Dylewski et al. demonstrated lower complication rates for robotic segmentectomy compared to robotic lobectomy [81]. Various other studies have found similar rates of complications and mortality between robotic segmentectomy and VATS. In another study regarding oncological outcomes of robotic segmentectomy, Nguyen et al. showed a 14% upstaging rate, 6% local or regional recurrence rate, and 73% lung cancer-specific survival for pathological stage I NSCLC. The authors concluded that robotic segmentectomy is a feasible and safe procedure for selected patients with decreased pulmonary function [82]. In a retrospective study by Xie et al. that analysed data from 215 patients who underwent atypical or anatomical segmentectomy by either RATS or conventional VATS, RATS was deemed safe and resulted in higher rates of lymph node dissection than conventional VATS, without an increase in postoperative complication rates [83]. Despite these promising findings, propensity-score matched studies or RCTs have yet to demonstrate these results.

## 4. Nodule Detection

The rise in detection of GGO and sub-centimetric pulmonary nodules combined with the increase in MIS (VATS/RATS) as a standard approach has created a need for optimal pre- and perioperative nodule detection techniques. For these smaller lesions, visual inspection of indirect signs, such as visceral pleura retraction, or manual palpation are insufficient [84]. Failure of nodule localization can lead to conversion to open surgery with some studies reporting rates up to 63%, especially for GGO and nodules < 10 mm in diameter or located >5 mm deep from the pleura [85]. Several strategies exist to approach this problem, such as CT-guided localization techniques using hook wire positioning or microcoil placement, preoperative marking with radio-labelled iodine-125 seeds, percutaneous or intravenous injection of liquid agents, and intraoperative ultrasound [27,28,29]. However, complications and practical issues have been reported for several of these techniques. CT-guided percutaneous fiducial marker placements can result in pneumothorax, pulmonary hematomas, or can be dislodged accidentally. For percutaneous liquid dyes, diffusion away from the nodule can limit their potential, especially in cases where the interval between labelling and thoracoscopic surgery is too long [85,86]. In more recent years, electromagnetic navigational bronchoschopy (ENB) has been gaining popularity has a guiding diagnostic and dye marking technique. With this technique, a virtual 3-dimensional (3D) model reconstruction of the patient’s lung is created using CT imaging. Subsequently, bronchoscopy is performed using an electromagnetic locatable guide with the patient lying on an electromagnetic board. The virtual 3D airway based on the preoperative CT scan is used to navigate the bronchoscope to the target lesion. A dye medium can then be injected close to the lesion. This hybrid technique significantly reduces dye diffusion and the occurrence of pneumothorax [87]. Several studies have reported diagnostic yields ranging between 38% up to 94% for small, peripheral lung nodules. The safety of this technique has also been shown in a number of studies [86]. 

## 5. Costs and Ergonomics of Robotic Surgery

In the last decade, the use of robotic systems in thoracic surgery has increased exponentially worldwide [21]. Even though RATS has been shown to provide many benefits compared to open surgery, the high costs of acquiring and maintaining these systems are not always mentioned in studies regarding RATS. Several cost analyses have been published in the last few years, with results varying widely across different areas and nations. Furthermore, large discrepancies exist between the results of these studies due to different definitions of cost [21]. A few studies have suggested that the total costs of RATS are comparable to open surgery due to the shortened hospital LOS and lower complication rates [88,89]. However, the majority of studies have shown that RATS is associated with higher costs than open surgery or VATS, mainly due to the costs of surgical robot system, more expensive intraoperative consumables, and longer operating room time [90]. However, a significant part of these cost analyses derive from studies based on early experiences with RATS. In a recent study by Heiden et al., cost-effectiveness analysis of RATS lobectomy was performed. Their results showed that RATS lobectomy is not cost effective at willingness-to-pay (WTP) thresholds of $50,000 and $100,000, only at a WTP threshold of $150 000. Nevertheless, if surgeons could perform as few as 1–3% more cases minimally invasively, then RATS lobectomy would become as cost effective as VATS [91]. However, more well-designed, prospective studies are necessary to assess this further. In addition, several different types of robotic systems have been introduced in recent years, creating more economic competition, which will likely result in lower prices for the future [21,49].

The increase in MIS in the past decades has offered clear benefits for patients compared to open surgery. However, MIS presents numerous ergonomic challenges to the surgeon. Surgeons are highly susceptible to work-related musculoskeletal disorders (MSDs), which are often overlooked by both practicing surgeons and surgical residents [92]. Laparoscopic/thoracoscopic procedures, generally involving extended periods of static postures and repetitive movements, can lead to injuries like carpal tunnel syndrome, wrist tendonitis, and thoracic outlet syndrome [93,94]. Furthermore, only a limited number of surgeons receive training in ergonomic techniques during surgery [94]. A number of studies have already compared the ergonomics of robotic surgery to laparoscopic surgery. While most of these studies have reported reduced discomfort and fatigue among surgeons using robotic surgery, they have relied heavily on subjective data using non-validated assessment tools [95,96]. Improved research tools are needed to identify surgical ergonomic risk factors and implement interventions that can prevent MSDs among surgeons.

## 6. Conclusions

With the rise in robotic surgery and the recent promising results regarding sublobar resections, it is clear that we have entered a new era of thoracic surgery. As CT-screening programs are increasingly implemented, increasing numbers of patients with small peripheral nodules that are eligible for surgery will present. In the near future patient-tailored “precision” surgery will become the norm. The future thoracic surgeon will need to have expertise in sublobar resections and complex segmentectomies. Even more so than now, close collaboration with pathologists and pulmonologists will be necessary to provide adequate surgical care. Furthermore, hybrid operating rooms with the possibility of CT-guided localization, ENB and fluorescence-guided surgery are necessary to increase efficiency during surgery for smaller nodules or GGO.

## Figures and Tables

**Table 1 cancers-15-04281-t001:** Overview of recent (<5 years old) meta-analyses and RCTs comparing VATS and RATS lobectomy/segmentectomy for NSCLC.

Study	Year	Type of Study	Number of Patients	Comparison	Outcomes	Results
Liang et al. [37]	2018	Meta-analysis (14 studies)	7438 (3239 RATS and 4199 VATS)	VATS vs. RATS lobectomy/segmentectomy	Perioperative and short-term outcomes	30-day mortality and conversion to open significantly lower in RATS group (*p* = 0.045 and *p* < 0.001, resp.). No other significant differences.
Guo et al. [36]	2019	Meta-analysis (14 studies)	20,948 (2553 RATS and 18,431 VATS)	VATS vs. RATS lobectomy	Perioperative and short-term outcomes	No signifiant difference between VATS and RATS regarding conversion rates, lymph node dissection, hospital LOS, surgical duration, chest drainage volume, PAL, and morbidity.
Hu et al. [48]	2020	Meta-analysis (32 studies)	6593 (2346 RATS, 2553 VATS, and 1694 open)	VATS vs. RATS lobectomy/segmentectomy	Perioperative and short-term outcomes	RATS had longer operative times and higher lymp node dissection rates compared to VATS. No other significant differences.
Wu et al. [49]	2020	Meta-analysis (25 studies)	50,404 patients (7135 RATS and 43,269 VATS)	VATS vs. RATS anatomical resections	Long- and short-term outcomes	RATS had a longer DFS compared to VATS (*p* = 0.03). OS showed a similar trend but was not statistically significant (*p* = 0.10). RATS showed a significantly lower 30-day mortality (*p* = 0.002). No significant difference was found in postoperative complications, conversion rate to open surgery, or lymph node upstaging.
Mao et al. [50]	2021	Meta-analysis (18 studies)	60,349 patients (8726 RATS and 51,623 VATS)	VATS vs. RATS lobectomy/segmentectomy	Perioperative and short-term outcomes	RATS had longer operative times (*p* < 0.001), lower postoperative complication rates after 2015 (*p* = 0.010), and improved lymph node dissection rates (*p* = 0.001) No other significant differences.
ROMAN Study (Veronesi et al.) [45]	2021	RCT	76 patients (39 RATS and 38 VATS)	VATS vs. RATS (bi)lobectomy/segmentectomy	Perioperative outcomes	RATS had improved lymph node dissection rates (*p* = 0.0002). No significant difference regarding perioperative complications, conversions, duration of surgery, or duration of hospital LOS.
Ma et al. [51]	2021	Meta-analysis (18 studies)	11,247 patients (5114 RATS and 6133 VATS)	VATS vs. RATS lobectomy/segmentectomy	Long- and short-term outcomes	No significant difference between RATS and VATS in operative time, mortality, OS, and DFS. Sensitivity analysis showed no significant regarding conversion rate, number of harvested lymph nodes and stations, and overall complications.
Zhang et al. [41]	2022	Meta-analysis (26 studies)	45,733 patients (14,271 RATS and 31,462 VATS)	VATS vs. RATS lobectomy/segmentectomy	Long- and short-term outcomes	RATS had less blood loss, a lower conversion rate to open, a shorter hospital LOS, more lymph node dissection, and better 5-year DFS compared to VATS group: No significant differences in operative time, any complications, tumor size, chest drain duration, R0 resection rate, lymph station, 5-year OS, and recurrence rate.
Jin et al. [46]	2022	RCT	320 patients (157 RATS and 163 VATS)	VATS vs. RATS lobectomy	Short-term outcomes	RATS had significantly higher number of harvested lymph nodes (*p* = 0.02) and stations (*p* < 0.001). Perioperative outcomes were comparable between the two groups, including the hospital LOS (*p* = 0.76) and the rate of postoperative complications (*p* = 0.45). No perioperative mortality occurred in either group.

DFS, disease-free survival; LOS, length of stay; NSCLC, non-small cell lung cancer; OS, overall survival; PAL, postoperative air leak; RATS, robotic-assisted thoracoscopic surgery; RCT, randomized controlled trial; VATS, video-assisted thoracoscopic surgery.

**Table 2 cancers-15-04281-t002:** Overview of ongoing RCTs regarding outcomes of segmentectomy in NSCLC.

Clinical Trial	Phase	Inclusion	Comparison	Estimated Enrollment (nr. of Patients)	Endpoint(s)
NCT02481661	III	cT1aN0M0 peripheral NSCLC	Lobectomy vs. anatomic segmentectomy	610	Primary: 5-year RFS Secondary: 5-year OS, retaining pulmonary function, and the rates of loco-regional and systemic recurrence
NCT02718365	NA	Stage IA NSCLC	Wedge resection vs. anatomic segmentectomy	1382	Primary: 5-year PFS Secondary: 3-year PFS, 5-year OS, retaining pulmonary function in the 1st year after surgery, 30-day morbidity, 30-day mortality, 10-year OS
NCT04944563	NA	Early-stage NSCLC ≤ 2 cm in the middle third of lung field.	Lobectomy vs. anatomic segmentectomy	1120	Primary: 5-year DFS Secondary: 3-year DFS, 5-year OS, retaining pulmonary function in the 1st year after surgery, 30-day morbidity, 30-day mortality
NCT02360761	III	Elderly patients with cT1N0M0 NSCLC	Sublobar resection vs. lobectomy	339	Primary: 3-year DFS Secondary: perioperative complications, 30-day mortality, hospital LOS, intubation time after surgery, 3-year OS, 3-year PFS, retaining pulmonary function 3 years postoperatively, percentage of VATS procedures, QoL scores
NCT03066297	NA	Clinical stage IA NSCLC	Wide wedge resection vs. segmentectomy vs. lobectomy	1000	Primary: 5-year DFS Secondary: 5-year OS, rate of locoregional recurence at 5 years, rate of systemic recurrence at 5 years, FEV1 (until 1 years postoperatively), DLco (until 1 years postoperatively), perioperative complications, C/T ration 2 months preoperatively, pathology subtype, incidence of LN metastasis
NCT02011997	III	Clinical stage IA NSCLC	VATS lobectomy vs. segmentectomy	500	Primary: 5-year RFS Secondary: 5-year OS, postoperative complications, pulmonary function (until 6 months postoperatively), QoL scores
NCT00499330	III	Clinical stage IA NSCLC	Lobectomy vs. segmentectomy	701	Primary: 7-year DFS Secondary: 7-year OS, rate of locoregional or systemic recurence at 7 years, FEV1 (at 6 months postoperatively)
NCT03108560	NA	cT1N0M0 NSCLC	Sublobar resection vs. lobectomy	600	Primary: 5-year OS Secondary: 5-year DFS, rate of locoregional or systemic recurence at 5 years, pulmonary function (until 2 years postoperatively), 30-day morbidity, 30-day mortality

DFS, disease-free survival; DLco, diffusion capacity of the lung for carbon monoxide; FEV1, forced expiratory volume in 1 s; LN, lymph node(s); LOS, length of stay; NA, not applicable; NSCLC, non-small cell lung cancer; OS, overall survival; PAL, postoperative air leak; PFS, progression-free survival; RATS, robotic-assisted thoracoscopic surgery; RCT, randomized controlled trial; RFS, recurrence-free survival; QoL, quality of life; VATS, video-assisted thoracoscopic surgery.

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
