# Peer review of "Minimally Invasive Surgery in Non-Small Cell Lung Cancer: Where Do We Stand?"

_cancers, 2023, doi:10.3390/cancers15174281_

Round 1
Reviewer 1 Report
Dear Editor and Authors,
It was my pleasure to evaluate this quite thorough and extensive review article by Dr. Berzenji and Professor Hendriks team from Department of Thoracic and Vascular Surgery, Antwerp University Hospital, University of Antwerp in Belgium. In this work the authors provide an overview of modern minimally invasive thoracic surgical techniques, especially focusing on robotic surgery its benefits and drawbacks.
This is a well written and well-presented review in clear language and easy to understand. However, it deals with significant controversial issues in modern thoracic surgery and questions which have not yet adequately been addressed. As such there needs to be a prudent and balanced approach. I have a number of issues I would like to bring up if I may! Specifically:
1. I am not certain that segmentectomy is becoming the new standard of surgical resection, certainly this is where we are headed but it is not clearly established yet. Please modify the statements made in the abstract and in the main text.
2. In lines 103-104 only 2 of the many possible and under investigation nodule locating techniques are listed. I suggest radiolabeling, wire/coil insertion and ultrasound are also added with appropriate references like they have done in section 4.
3. It is obvious the authors are pre-minimally invasive technique, which is maybe the sign of the times and this is quite evident from the way they are writing and presenting the literature. Not that this reviewer is against minimally invasive surgery (quite to the contrary) however if one wishes to produce a beneficial to the community work we need to be balanced as our ancient ancestors used to say!! For example, the authors mention reduced length of stay but fail to mention what that translates into. The difference as reported in the literature is 1.3 days, lets say maximum 2 so in reality this clinical benefit is not so significant!! In terms of recovery, yes a thoracotomy has more pain and slower recovery but as the literature shows there is no oncological difference and change in long term survival between the two approaches (this is the bottom line!!). And this may not be fully true because of the issue with completeness of lymph node dissection and staging with MIS/VATS!! These issues need to be presented in a balanced as mentioned approach and let the readers make up their mind when presented with all the evidence. Let us not spoon feed them what we feel is right!!
4. Another parameter that is concerning and the authors should try to mention/address is the training required to be able to perform VATS and RATS procedures and more importantly how MIS is producing a generation of surgeons who could not function if you take their nifty toys away!! During the financial crisis and fiscal austerity, we were faced with a significant shortage of staplers and surgical disposables. We had to therefore resort to old conventional, traditional thoracic surgery i.e. ligating vessels using suture ties and suturing parenchyma by hand, and in truth it did not make that much of a difference in terms of outcome as we have reported! Therefore, this issue needs to be mentioned in this review and analyzed further!
5. Again in lines 240 to 241 the authors show their bias “In the near future, uniportal RATS (UniRATS) will likely become a standard approach as well” when in truth no robust data exist to demonstrate that uniportal VATS is significant better than biportal VATS or three port VATS!! Please rephrase and avoid such exclusive and controversial statements!! Does this mean that all we surgeons who perform two and three port VATS are not practicing surgery correctly and we are liable for malpractice since it is not “the gold standard”?? See how someone could misconstrued what you say!!
6. The whole section about immunotherapy and its response rates seems superfluous and has no impact on surgical technique. The data out there are still lacking and mentioning this issue in relationship to surgical outcomes needs to be measured and prudent!
7. The way that the lower relapse free survival is reported in anatomical segmentectomies is problematic and it feels like it is been downplayed. For example the reporting of the Japanese JCOG0802 trial is reported in lines 282-283!! Please rephrase and edit this whole section!
8. This whole mention about cost effectiveness of RATS is also an issue. The authors concede that RATS is a more expensive technique due to the inherent initial purchase cost, the cost of disposables, the length of surgical procedures ect but then attempt to whitewash over the issue by not presenting clearly the evidence and concluding “However, a significant part of these cost analyses derive from studies based on early experiences with RATS and are outdated.”!! We need a more balanced and extensive review of the cost because it is a major issue in the propagation of RATS!!
In conclusion this is a decent attempt at a review of a quite controversial issue (or issues) and in general I am positively inclined to agree that a comprehensive review has a place out there and can be useful to the surgical community. HOWEVER, it needs to be balanced and objective so that we do not drive the readers with our own biases!! I therefore recommend that the authors revise their work before it is re-considered for publication. Overall this shows commitment and a lot of effort on behalf of the author so, good job and I am awaiting your revision!
English is good.
Author Response
We would like to thank the reviewer for the insightful comments and addressing the need for nuance in our manuscript. Below are the point-by-point responses to the comments.
1. I am not certain that segmentectomy is becoming the new standard of surgical resection, certainly this is where we are headed but it is not clearly established yet. Please modify the statements made in the abstract and in the main text.
Response 1: The statements have been modified and nuanced.
2. In lines 103-104 only 2 of the many possible and under investigation nodule locating techniques are listed. I suggest radiolabeling, wire/coil insertion and ultrasound are also added with appropriate references like they have done in section 4.
Response 2: Radio-guided localization, wire/coil implantation and ultrasonography have been mentioned together with the appropriate references (line 102-105).
3. It is obvious the authors are pre-minimally invasive technique, which is maybe the sign of the times and this is quite evident from the way they are writing and presenting the literature. Not that this reviewer is against minimally invasive surgery (quite to the contrary) however if one wishes to produce a beneficial to the community work we need to be balanced as our ancient ancestors used to say!! For example, the authors mention reduced length of stay but fail to mention what that translates into. The difference as reported in the literature is 1.3 days, lets say maximum 2 so in reality this clinical benefit is not so significant!! In terms of recovery, yes a thoracotomy has more pain and slower recovery but as the literature shows there is no oncological difference and change in long term survival between the two approaches (this is the bottom line!!). And this may not be fully true because of the issue with completeness of lymph node dissection and staging with MIS/VATS!! These issues need to be presented in a balanced as mentioned approach and let the readers make up their mind when presented with all the evidence. Let us not spoon feed them what we feel is right!!
Response 3: Thank you for your insightful comment. You are right that our statements needed to be balanced against the clinical reality. We have added a paragraph (line 290-306) in which all these issues are addressed and discussed.
4. Another parameter that is concerning and the authors should try to mention/address is the training required to be able to perform VATS and RATS procedures and more importantly how MIS is producing a generation of surgeons who could not function if you take their nifty toys away!! During the financial crisis and fiscal austerity, we were faced with a significant shortage of staplers and surgical disposables. We had to therefore resort to old conventional, traditional thoracic surgery i.e. ligating vessels using suture ties and suturing parenchyma by hand, and in truth it did not make that much of a difference in terms of outcome as we have reported! Therefore, this issue needs to be mentioned in this review and analyzed further!
Response 4: This issue has been addressed as well in the new paragraph (line 290-306)
5. Again in lines 240 to 241 the authors show their bias “In the near future, uniportal RATS (UniRATS) will likely become a standard approach as well” when in truth no robust data exist to demonstrate that uniportal VATS is significant better than biportal VATS or three port VATS!! Please rephrase and avoid such exclusive and controversial statements!! Does this mean that all we surgeons who perform two and three port VATS are not practicing surgery correctly and we are liable for malpractice since it is not “the gold standard”?? See how someone could misconstrued what you say!!
Response 5: Thank you for showing how our text could be misconstrued. We are certainly not insinuating that multiportal VATS is inferior to uniportal VATS. It is very true that there is no robust data on outcomes of uniVATS/uniRATS. We were trying to demonstrate the rapidly developing technical modalities in thoracic surgery. We have adapted the statement to make this more clear.
6. The whole section about immunotherapy and its response rates seems superfluous and has no impact on surgical technique. The data out there are still lacking and mentioning this issue in relationship to surgical outcomes needs to be measured and prudent!
Response 6: We mentioned immunotherapy because of its increasing importance in the treatment of NSCLC. However, it is true that data on its impact in the surgical treatment is lacking. We have addressed this issue in our text (line 287-289).
7. The way that the lower relapse free survival is reported in anatomical segmentectomies is problematic and it feels like it is been downplayed. For example the reporting of the Japanese JCOG0802 trial is reported in lines 282-283!! Please rephrase and edit this whole section!
Response 7: We have extended the section on segmentectomy vs. lobectomy and included more of the results of the JCOG0802 trial (lines 321-354).
8. This whole mention about cost effectiveness of RATS is also an issue. The authors concede that RATS is a more expensive technique due to the inherent initial purchase cost, the cost of disposables, the length of surgical procedures ect but then attempt to whitewash over the issue by not presenting clearly the evidence and concluding “However, a significant part of these cost analyses derive from studies based on early experiences with RATS and are outdated.”!! We need a more balanced and extensive review of the cost because it is a major issue in the propagation of RATS!!
Response 8: We have addressed the issue of costs and added some more data (and nuance) on this matter in the section of cost effectiveness of RATS (lines 417-426).
Reviewer 2 Report
In this study, the authors reviewed the current state of minimally invasive surgery (MIS) for nonsmall cell lung cancer (NSCLC). They particularly talked focusing on RATS and sublobar resection. This manuscript is well organized and written, and could be accepted. But I have some comments listed below.
1. They evaluated the goodness of RATS compared to VATS or open about short-term and oncological outcomes, and concluded that RATS could gain popularity as a MIS alternative to VATS even although the high costs, which is the most serious disadvantage. I can agree their conclusions. However, their speculation could be demonstrated from the surgeon’s perspective. I think they should discuss about the patient’s perspective much more such as postoperative QOL (line 180-182) and others. Because I think RATS might not reach a highly recommended technique for NSCLC patients. They should address.
2. Table 2: Sublobar resection could be less invasive and an alternative method compared to lobectomy for small sized NSCLC from the results of long-term survival outcome even though there is a higher rate of loco-regional recurrence. I can agree. In table 2, the authors showed some ongoing studies about sublobar resection vs. lobectomy. The aim of these almost all studies are same, and their message could not be so impressive for readers. There were some important meta analyses and RCT trials published so far. Preferably, it should be better to show their information in Table 2. They should think about it.
3. Line 330-354: To detect GGO and sub-centimetric pulmonary nodules accurately is so important in case of sublobar resection such as wide wedge resection. There are so many detection methods reported so far, and the authors introduced some reported techniques. However, they focused on minimally invasive surgery from the point of views of surgical techniques (RATS…) and resection volume (sublobar resection…). For me, there is no need to talk about the marking methods for small-sized pulmonary nodules as a special section in this review although I can understand its importance.
Author Response
We would like to thank the reviewer for the kind words and the insightful comments. Below are our responses.
1. They evaluated the goodness of RATS compared to VATS or open about short-term and oncological outcomes, and concluded that RATS could gain popularity as a MIS alternative to VATS even although the high costs, which is the most serious disadvantage. I can agree their conclusions. However, their speculation could be demonstrated from the surgeon’s perspective. I think they should discuss about the patient’s perspective much more such as postoperative QOL (line 180-182) and others. Because I think RATS might not reach a highly recommended technique for NSCLC patients. They should address.
Response 1: Thank you for the insight, we have added some lines addressing this matter (lines 301-306)
2. Table 2: Sublobar resection could be less invasive and an alternative method compared to lobectomy for small sized NSCLC from the results of long-term survival outcome even though there is a higher rate of loco-regional recurrence. I can agree. In table 2, the authors showed some ongoing studies about sublobar resection vs. lobectomy. The aim of these almost all studies are same, and their message could not be so impressive for readers. There were some important meta analyses and RCT trials published so far. Preferably, it should be better to show their information in Table 2. They should think about it.
Response 2: Our aim was to show the ongoing trials in Table 2. However, we have added some more data on the JCOG0802 and the Altorki trial in lines 321-354.
3. Line 330-354: To detect GGO and sub-centimetric pulmonary nodules accurately is so important in case of sublobar resection such as wide wedge resection. There are so many detection methods reported so far, and the authors introduced some reported techniques. However, they focused on minimally invasive surgery from the point of views of surgical techniques (RATS…) and resection volume (sublobar resection…). For me, there is no need to talk about the marking methods for small-sized pulmonary nodules as a special section in this review although I can understand its importance.
Response 3: We think that localization techniques will become increasingly important in the era of of sublobar resections, therefore we included a section on this matter. We have mentioned some additional techniques to make this section more comprehensive.
Round 2
Reviewer 1 Report
Dear Editor and Authors,
It was really a pleasure to re-review the revised manuscript by Dr. Berzenji and his colleagues. The authors have addressed all the issues I raised to a very good level and I thank them for that! I hope my commentary has been constructive and I do feel the corrections have improved the manuscript. Certainly a more ballanced view is now provided which was one of the issues I had identified in my primary review.
Therefore, I am now more than happy to recommend the acceptance of this work. Congradulations to the authors for a very nice work.
Kind regars to all.
Only minor editing is required which can be done at proofreading!